environmental science/bioengineering

sediment, denitrifying bacteria, denitrification, cultivation

**Author for correspondence:**
Jun Li
e-mail: bjutlijun@sina.com

# Cultivating river sediments into efficient denitrifying sludge for treating municipal wastewater

Liangang Hou, Jun Li, Zhaoming Zheng, Qi Sun, Yitao Liu and Kai Zhang

College of Architecture and Civil Engineering, Beijing University of Technology, Beijing 100124, People's Republic of China

(iD) JL, 0000-0002-4392-3919

The river sediment contains a lot of pollutants in many cases, and needs to be treated appropriately for the restoration of water environments. In this study, a novel method was developed to convert river sediment into denitrifying sludge in a sequencing batch reactor (SBR). The river sediment was added into the reactor daily and the hydraulic retention time (HRT) of the reactor was gradually reduced from 8 to 4 h. The reactor achieved in the $NO_3^-$-N removal efficiency of 85% with the $NO_3^-$-N removal rate of 0.27 kg N m$^{-3}$ d$^{-1}$. Response surface analysis represents that nitrate removal was affected mainly by HRT, followed by sediment addition. The denitrifying sludge achieved the highest activity with the following conditions: $NO_3^-$-N 50 mg l$^{-1}$, HRT 6 h and adding 6 ml river sediments to 1 l wastewater of reactor per day. As a result, the cultivated denitrifying sludge could remove 80% $NO_3^-$-N for real municipal wastewater, and the high-throughput sequence analysis indicated that major denitrifying bacteria genera and the relative abundance in the cultivated denitrifying sludge were *Diaphorobacter* (33.82%) and *Paracoccus* (24.49%). The river sediments cultivating method in this report can not only obtain denitrifying sludge, but also make use of sediment resources, which has great application potential.

## 1. Introduction

Sediments are formed by the accumulation of various substances in a wide range of spaces of water ecosystem over long periods of time [1,2]. The river sediments have a high moisture content and complex composition [3,4], and the toxic or harmful substances in some rivers have significant adverse impact on the quality of the water environment [5]. At the same time, the sediments contain a

large amount of organic and nutrients [6], such as carbohydrates, N, P, K, etc. [7,8], which has great economic value and can be effectively used [9]. At present, the disposal of sediments mainly includes dredging [10] and landfill [11–13] which can easily cause secondary pollution [14,15] and are high in cost [16–18]. Therefore, advanced and environment-friendly methods that can effectively dispose of sediments are urgently needed.

The traditional denitrification processes in most wastewater treatment plants (WWTPs) usually require a large number of denitrifying bacteria [19], sludge with denitrifying bacteria [20,21] is usually used for biological denitrification [22], as a result, great amounts of denitrifying sludge is required [23,24]. Sludge used in WWTPs is generally obtained by long-term cultivation or sludge recirculation [25–27]. However, longer cultivation time or sludge recirculation requires energy [28,29], which increases the cost [30]. Therefore, developing a new method for cultivating denitrifying sludge with low cost and energy consumption is a necessary choice for WWTPs [31].

Based on the abundant available organic matter and diverse bacteria in river sediments [32–34], this study reports a novel method of cultivating denitrifying sludge with sediments, which not only provides sludge for the denitrification process of WWTPs, but also seeks an economic resource utilization way to effectively treat sediments. In this study, denitrifying sludge cultivated by river sediments was used for the treatment of municipal wastewater to test its performance. The denitrification performance of the cultured sludge was investigated and its optimal cultivation conditions were explored.

# 2. Material and methods

## 2.1. Sediment and wastewater

The river sediment used in this study was taken from Xiaotaihou River in Beijing, China. The river sediment was sieved by using 0.5 mm mesh sieve. Glucose was added into the river sediment, and the concentration of the glucose in the mixture was $5 \, g \, l^{-1}$. The concentration of the mixed liquor volatile suspended solids (MLVSS) of the sediment was about $8500 \, mg \, l^{-1}$.

The feeding medium used in this experiment included synthetic wastewater and municipal wastewater. In the cultivation process, synthetic wastewater was used as the influent, with different concentrations of nitrate and COD at different stages in the form of $NaNO_3$ and glucose, respectively. The composition of the mineral medium was: $KH_2PO_4$, $CaCl_2 \cdot 2H_2O$, $MgSO_4 \cdot 7H_2O$, $NaHCO_3$, $FeSO_4 \cdot 7H_2O$ and $1 \, ml \, l^{-1}$ (v/v) trace element solution. The composition of the trace element solution was based on a previous study [35]. The municipal sewage was collected from effluent of aeration tank, one of the WWTPs in Liaoning Province, China. The major characteristics of the municipal wastewater include: $NO_3^-N$ $25 \pm 2 \, mg \, l^{-1}$, $NH_4^+N$ less than $0.1 \, mg \, l^{-1}$, $NO_2^-N$ $0.2 \pm 0.1 \, mg \, l^{-1}$, chemical oxygen demand (COD) $50 \pm 5 \, mg \, l^{-1}$. The pH value of the municipal sewage fluctuated in the range of 7.6–7.7 and dissolved oxygen (DO) $< 0.5 \, mg \, l^{-1}$.

## 2.2. Experimental set-up

The schematic of the laboratory-scale sediment cultivating bioreactor is shown in figure 1. The bioreactor was made of a plexiglas cylinder of 350 mm in height and 140 mm in diameter, and with an operating volume of 5.0 l. A stirrer was used to prevent sediment from settling and the constant temperature water bath maintained the temperature between 28°C and 30°C that ensured the microorganisms in the reactor had a good activity. In total, 1.8 l synthetic water and 200 ml sediments were added into the reactor at the beginning, with the MLVSS about $850 \, mg \, l^{-1}$. The reactor was operated for 24 h per cycle during the 15 days operation. The operation stage of each cycle included four stages: influent for 10 min, reaction for 23 h (stirring slowly for 15 min, setting for 5 min, cycle operation), sludge precipitation for 40 min and 10 min of decanting. Hydraulic retention time (HRT) was 24 h, and running for 15 days. The change of $NO_3^-N$ in effluent is shown in figure 2, indicating that denitrifying bacteria appeared in sediments and the reactor was successfully started.

During the operation, the amount of added sediment was gradually increased and the HRT was shortened gradually. After the start-up of the reactor, synthetic water containing $20 \, mg \, l^{-1}$ $NO_3^-N$ and $105 \, mg \, l^{-1}$ COD was used as feeding water, and HRT = 12 h (10 min feeding, 11 h reaction (stirring slowly for 15 min, setting for 5 min, cycle operation), sludge precipitation for 40 min, 10 min discharging). The reactor was operated stably for 5 days under sequential batch to enrich denitrifying bacteria.

HRT was set to 8 h (the reaction time was adjusted to 7 h, and the others remained unchanged) from day 21 to day 35, and 20 ml sediment ($4 \, ml \, l^{-1}$, v/v) was added into the 5.0 l reactor every day; simultaneously, the

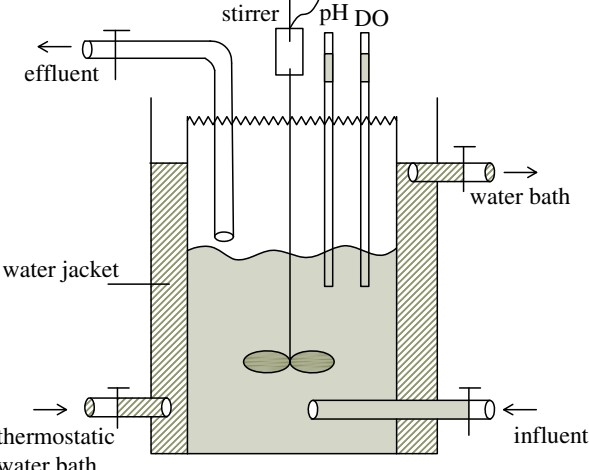

**Figure 1.** Schematic diagram of the reactor.

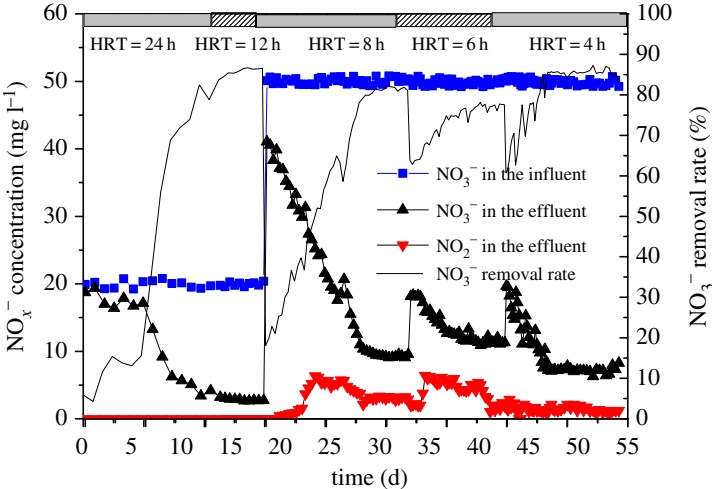

**Figure 2.** Changes of nitrogen form during cultivation process in effluent and influent.

water increased accordingly for keeping the moisture content maintained at 20%, and $NO_3^-$-N in influent increased to 50 mg l$^{-1}$. HRT was changed to 6 h (the reaction time was adjusted to 5 h) from the 36th day to the 45th, and 30 ml sediment (6 mg l$^{-1}$, v/v) was added into the reactor per day, and the others remained unchanged. In a similar way, HRT was maintained at 4 h (the reaction time was adjusted to 3 h) from the 46th day to the 50th, and 40 ml sediment (8 ml l$^{-1}$, v/v) was added into the reactor daily, and the others remained unchanged. When the reactor was in operation for 50 days, 1.0 l sediment was added into the reactor. The mixture of mud and water was about 5.0 l, and MLVSS was about 1700 mg l$^{-1}$. The denitrification tended to be stable when the operation continued for 5–7 days after the completion of sediment cultivation.

## 2.3. Chemical analysis

$NO_3^-$-N and $NO_2^-$-N were analysed according to the standard methods [36]. Before analysis of the above parameters in liquid, samples were membrane filtered (0.45 μm). The temperature, pH and dissolved oxygen (DO) were measured near the midway of the reactor by using a WTW analyser (Multi 3620IDS, Germany).

At the end of the experiment, the sediment in the sequencing batch reactor (SBR) was collected for scanning electron microscopy (SEM) analysis, samples were fixed in phosphate buffer (400 mM, pH = 7.4) with 4% glutaraldehyde, and rinsed in phosphate buffer containing saccharose (400 mM). They were dehydrated by immersion in solutions with increasing concentrations of acetate (50%, 70%, 100%), then in acetone and hexamethyldisilazane (HMDS) (50 : 50), and finally in 100% HMDS. The last batch of HMDS was dried until complete evaporation [37].

## 2.4. Community analysis

The sediment in the SBR was collected at the beginning and end of cultivation for community analysis. We extracted DNA using the FastDNA SPIN Kit for Soil, following the manufacturer's protocol, characterized microbial species by high-throughput sequencing [38]. The universal bacteria primers which incorporated Miseq platformic the V3–V4 hypervariable regions were applied to amplify the extracted DNA. The final sequences of the primers were 341F(CCTACGGGNGGCWGCAG),805R(GACTACHVGGGTATCTA ATCC). The PCR products were sequenced on the Miseq $2 \times 300$ bp pyrosequencing platform by Sangon Biotech, Shanghai, China.

## 2.5. Response surface analysis

In order to obtain the optimum sediment cultivation conditions, the response surface methodology (RSM) was used to optimize the process parameters. The optimized parameters are: HRT (h), sediment addition (SA, ml l$^{-1}$), nitrate concentration (NC, mg l$^{-1}$) and nitrate removal rate (NRR, %) as response values for RSM experiment.

# 3. Results and discussion

## 3.1. Changes of nitrogen form during cultivation process

The main purpose of the start-up phase is to enrich the denitrifying bacteria naturally existing in the sediment, so that the original small amount denitrifying bacteria becomes the dominant flora under the suitable growth conditions provided for them. The concentration of $NO_3^-N$ and $NO_2^-N$ in the influent and effluent during the experiment were measured, and the activity of the denitrifying bacteria was evaluated by the change of the NC in the water.

As shown in figure 2, in the start-up phase of the reactor, $NO_3^-N$ starts decreasing significantly from day 8 with the influent $NO_3^-N$ of 20 mg l$^{-1}$, indicating that the denitrifying bacteria in the reactor proliferated and denitrification occurred. After the reactor started, the denitrifying bacteria in the reactor are provided with more substrates by shortening the HRT to further enrich and cultivate. The effluent $NO_3^-N$ remained around 2.7 mg l$^{-1}$ with a $NO_3^-N$ removal efficiency of 86.5% when the reactor was operated from day 18 to 20, indicating that the denitrification in the reactor is relatively complete, moreover, the denitrifying bacteria have become the dominant species.

The purpose of microbial cultivation is to select and induce the mixed microflora, so that the microorganism with the activity of degrading pollutants becomes the dominant microflora [39]. On day 21, the microbial cultivation stage was started, and the cultivation method that the river sediment was added into the reactor daily with the gradually shortening HRT was carried out, and simultaneously increasing influent $NO_3^-N$ to 50 mg l$^{-1}$.

It can be seen from figure 2, under different hydraulic retention times in the cultivation stage that the $NO_3^-N$ concentrations in the effluent are 9.53, 11.31, 7.35 mg l$^{-1}$, respectively, with the removal rates of 81.08%, 77.57% and 85.21% corresponding.

When the cultivation process was completed for 3–5 days, the NC in the effluent was maintained at 7.2 mg l$^{-1}$ and the NRR was about 85.5% and 0.27 kg N m$^{-3}$ d$^{-1}$, indicating that the sediment has been cultivated into denitrifying sludge successfully. It was reported [40,41] that nitrite and its incorporation form free nitrous acid caused inhibition to a broad community of microbes. Due to the influence of HRT in the cultivation stage, denitrification was incomplete and a small accumulation of $NO_3^-N$ was observed in this study. The concentration of nitrite in the bioreactor was low, and the highest accumulation concentration of $NO_3^-N$ was 6.33 mg l$^{-1}$ (figure 2), so the concentration of free nitrous acid was very low, which has a little inhibitory effect on microbes. In short, according to the effluent nitrate and its removal efficiency, it can be concluded that the cultivation method can cultivate the sediment into denitrification sludge which has a highly efficient denitrification performance.

## 3.2. Optimal cultivation conditions

The results of the experiments are shown in table 1, and the adequacy of this model was evaluated through analysis of variance (ANOVA; table 2).

**Table 1.** Design and results of RSA experiments.

| no. | HRT | SA | NC | NRR | no. | HRT | SA | NC | NRR |
|-----|-----|-----|-----|------|-----|-----|-----|-----|------|
| 1 | 6 | 4 | 80 | 80.38 | 10 | 6 | 6 | 50 | 85.5 |
| 2 | 6 | 6 | 50 | 85.5 | 11 | 6 | 6 | 50 | 85.5 |
| 3 | 6 | 6 | 50 | 85.5 | 12 | 8 | 6 | 80 | 86.25 |
| 4 | 8 | 8 | 50 | 85.75 | 13 | 4 | 4 | 50 | 77.65 |
| 5 | 8 | 4 | 50 | 83.14 | 14 | 8 | 6 | 20 | 85.91 |
| 6 | 6 | 6 | 50 | 85.5 | 15 | 6 | 8 | 80 | 83.07 |
| 7 | 4 | 8 | 50 | 80.18 | 16 | 4 | 6 | 80 | 77.73 |
| 8 | 4 | 6 | 20 | 77.72 | 17 | 6 | 4 | 20 | 76.69 |
| 9 | 6 | 8 | 20 | 80.18 | | | | | |

**Table 2.** Response surface regression model analysis of variance test. Note: $R^2 = 0.9551$, $R_{adj} = 0.8975$; the difference is significant ($p < 0.05$), the difference is highly significant ($p < 0.01$), the difference was extremely significant ($p < 0.001$).

| source | sum of squares | d.f. | mean square | $F$-value | $p$-value Prob > $F$ |
|--------|----------------|------|-------------|-----------|----------------------|
| model | 191.14 | 9 | 21.24 | 16.56 | 0.0006 |
| A-HRT | 96.4 | 1 | 96.4 | 75.17 | <0.0001 |
| B-SA | 16.02 | 1 | 16.02 | 12.49 | 0.0095 |
| C-NC | 6 | 1 | 6 | 4.68 | 0.0673 |
| AB | 0.0016 | 1 | 0.0016 | 0.001248 | 0.9728 |
| AC | 0.027 | 1 | 0.027 | 0.021 | 0.8883 |
| BC | 0.16 | 1 | 0.16 | 0.12 | 0.7343 |
| $A^2$ | 4.2 | 1 | 4.2 | 3.27 | 0.1133 |
| $B^2$ | 33.51 | 1 | 33.51 | 26.13 | 0.0014 |
| $C^2$ | 28.44 | 1 | 28.44 | 22.17 | 0.0022 |
| residual | 8.98 | 7 | 1.28 | | |
| lack of fit | 8.98 | 3 | 2.99 | | |
| pure error | 0 | 4 | 0 | | |
| cor total | 200.12 | 16 | | | |

Multivariate regression analysis of the data (table 1) is conducted using the response surface software design expert 8.0.6. Equation (3.1) gives the regression equation of the $NO_3^- N$ removal rate during the cultivation process.

$$\begin{aligned} NRR = {} & 27.39125 + 4.63313 \times HRT + 9.30792 \times SA + 0.32937 \\ & \times NC + 0.005 \times HRT \times SA + 0.001375 \times HRT \times NC - 0.00333333 \\ & \times SA \times NC - 0.24969 \times HRT^2 - 0.70531 \times SA^2 - 0.0028875 \times NC^2. \end{aligned} \quad (3.1)$$

As shown in table 2, analysis of variance indicated that the model had better regression effects ($p = 0.0006$) and higher significance ($F = 16.56$). The difference of HRT ($p < 0.0001$) in the first term of the equation is highly significant, indicating that HRT has the greatest influence on $NO_3^- N$ removal; SA ($p < 0.05$) was a significant item showing that the amount of sediments added has impact on the removal rate of $NO_3^- N$. NC ($p > 0.05$) was not significant, indicating that the effect of NC was not significant in the experimental set-up. The influence of $B^2$ in the squared term is significant, indicating that the effect of experimental factors on the response value does not have a simple linear relationship, and the squared term has a greater influence on the response value. The decision coefficient $R^2 = 0.9551$ and the correction coefficient $R_{adj} = 0.8975$ indicate that the regression model fits

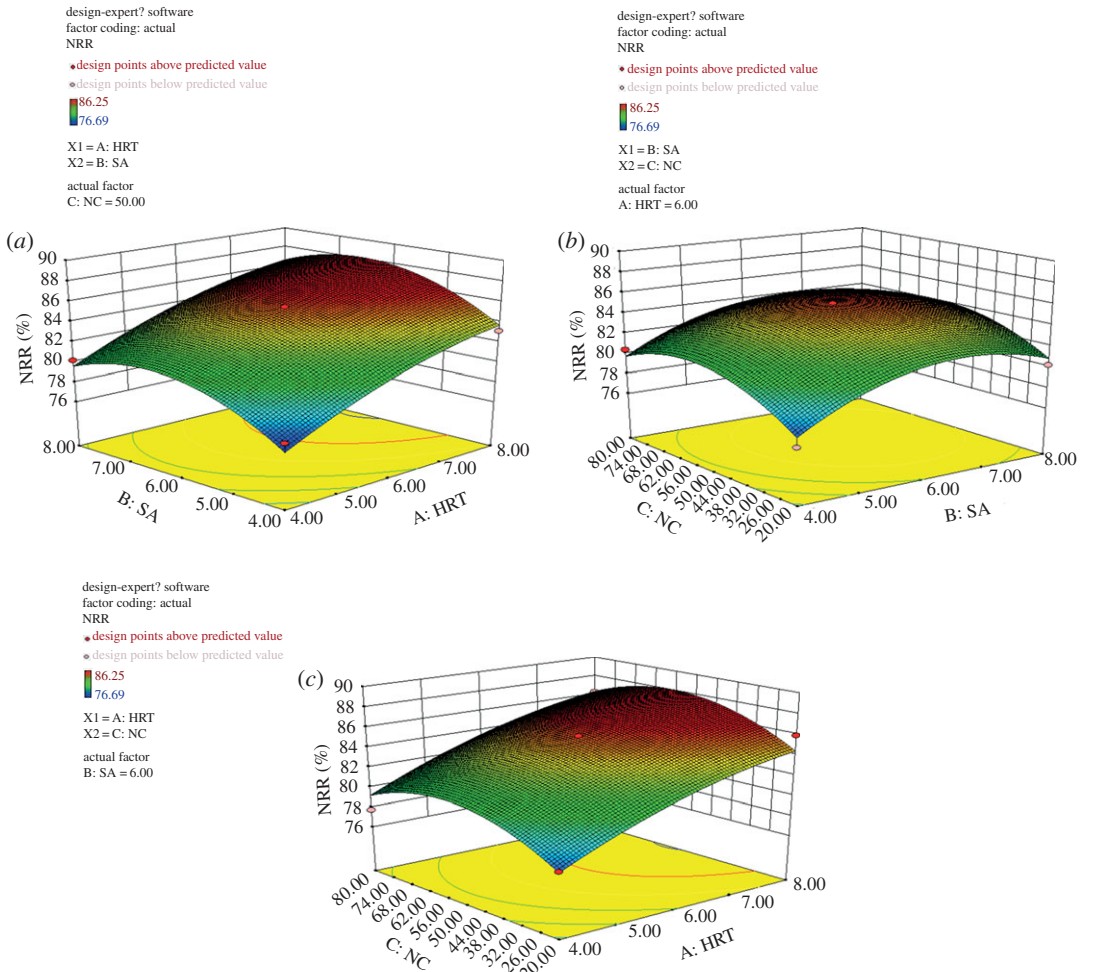

**Figure 3.** Response surface. (*a*) SA and HRT. (*b*) SA and NC. (*c*) NC and HRT.

well with the actual situation and can be used to optimize the $NO_3^-N$ removal during the cultivation process.

It can be seen from the value of $p$ that the influence of selected factors on the $NO_3^-N$ removal is HRT > SA > NC. The cor total is 200.12 > 4, indicating that the model works well [42]. Based on the regression equation obtained by regression model variance analysis, the response surface of HRT, SA and NRR was made by software (figure 3). When the third factor is at zero level, each graph can reflect the effect of the interaction of the other two factors on the $NO_3^-N$ removal rate.

It can be seen from figure 3*a* that when SA is constant, NRR increases with the increase of HRT, and the interaction between SA and HRT is obvious, and HRT has a significant effect on NRR. From figure 3*b*, when the NC is constant, the NRR increases first and then decreases with the increase of the SA amount, and the effect of SA on the NRR is more obvious. From figure 3*c*, when HRT is constant, the NRR increases first and then decreases with the increase of NC, but the interaction between the two is not obvious. The changes shown in figure 3 are consistent with the experimental conclusions in the single factor experiment. According to the established mathematical model after parameter optimization analysis, the denitrifying sludge reached the highest activity with the following conditions: $NO_3^-N$ 50 mg l$^{-1}$, HRT 6 h, and adding 6 ml river sediment to 1 l wastewater of reactor per day, and the nitrate removal efficiency at this time could reach 85.5%.

## 3.3. Microbial community analysis

The sediments which were stably operated for 2 days after the completion of cultivation were observed by SEM for bacterial morphology. A large number of dense bacterial micelles and pores were observed in the sludge from the SEM image (figure 4). It can be seen that the bacterial micelles in the sediments after cultivation are mainly rod-shaped cocci or Brevibacterium, and have a width of about 0.5–1.0 µm. High-

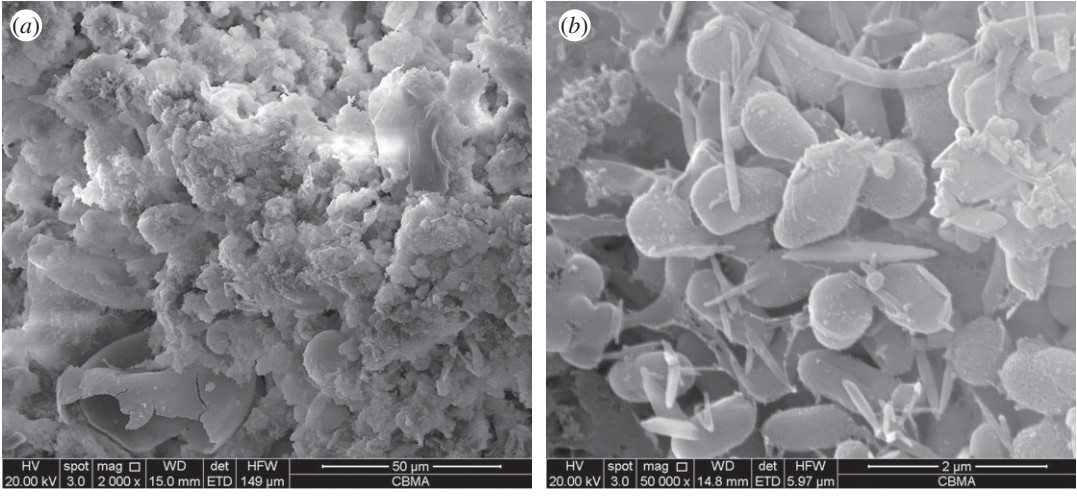

**Figure 4.** Scanning electron microscopy (SEM) images of the sediments, (*a*) (×2000) bar length: 50 μm; (*b*) (×50000) bar length: 2 μm.

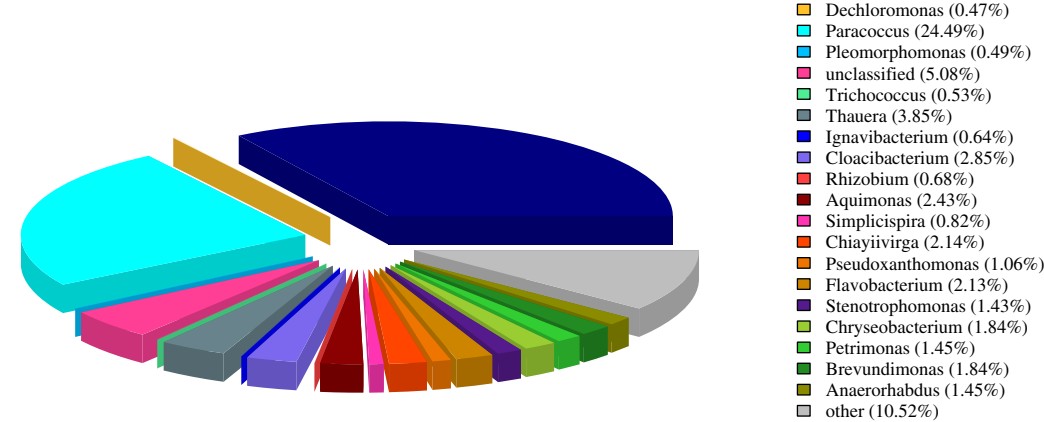

- Diaphorobacter (33.82%)
- Dechloromonas (0.47%)
- Paracoccus (24.49%)
- Pleomorphomonas (0.49%)
- unclassified (5.08%)
- Trichococcus (0.53%)
- Thauera (3.85%)
- Ignavibacterium (0.64%)
- Cloacibacterium (2.85%)
- Rhizobium (0.68%)
- Aquimonas (2.43%)
- Simplicispira (0.82%)
- Chiayiivirga (2.14%)
- Pseudoxanthomonas (1.06%)
- Flavobacterium (2.13%)
- Stenotrophomonas (1.43%)
- Chryseobacterium (1.84%)
- Petrimonas (1.45%)
- Brevundimonas (1.84%)
- Anaerorhabdus (1.45%)
- other (10.52%)

**Figure 5.** High-throughput sequencing analysis of the sediment after cultivation.

throughput sequencing was conducted to analyse the flora structure in the sediments at the end of the cultivation process, the main genera of denitrifying bacteria and the relative abundance in the cultivated denitrifying sludge were *Diaphorobacter* (33.82%) and *Paracoccus* (24.49%) (figure 5). They belong to the genus of denitrifying bacteria [43,44], which can convert $NO_3^-N$ into $N_2$ in an anaerobic environment. High-throughput sequencing indicates that the river sediments have been cultivated into denitrifying sludge.

## 3.4. Municipal wastewater denitrification

After the denitrification sludge was successfully cultivated, denitrifying nitrogen removal of municipal wastewater from aeration tank effluent was carried out in the SBR. The effective volume of the SBR was 5 l, and 750 ml sludge was inoculated with a filling ratio of 15% (v/v). The temperature was controlled by a water bath at $30 \pm 2°C$ with intermittent stirring, and the HRT is 6.5 ~ 4.5 h. According to previous studies, the denitrification process needs to consume carbon sources, the theoretical C/N ratio for complete biological denitrification is 2.86, and C/N should be maintained above 5.0 in practical applications. Since the $NO_3^-N$ concentration and the COD concentration of the influent was about $25 \pm 2$ and $50 \pm 5$ mg l$^{-1}$, respectively, the carbon source cannot meet the denitrification demand when the municipal wastewater is influent, so certain amount of glucose was added to maintain the influent COD at 130 ~ 150 mg l$^{-1}$.

Figure 6 shows the influent and effluent concentration of the reactor treating the municipal wastewater from the aeration tank of the WWTPs with the cultivated denitrifying sludge. After stable operation under

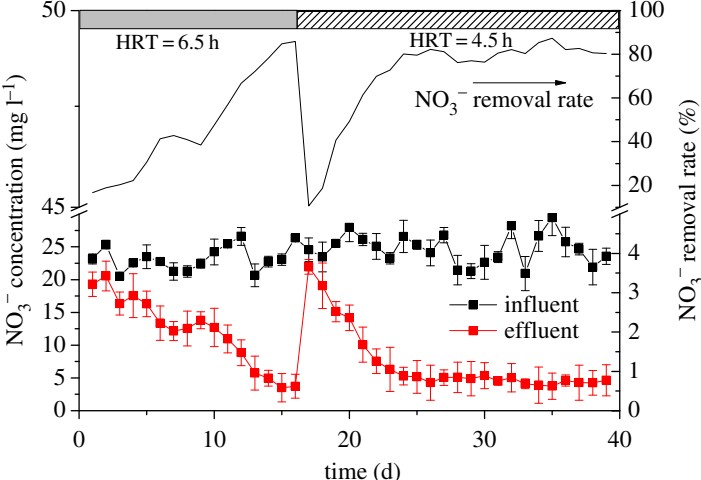

**Figure 6.** Changes of $NO_3^-$-N in municipal wastewater treatment.

the condition of HRT of 6 h, the effluent $NO_3^-$-N concentration was maintained at 3.5–3.7 mg $l^{-1}$ with the removal rate of 85%. When the HRT is adjusted to 4 h, the $NO_3^-$-N in effluent is about 4.2–4.6 mg $l^{-1}$, and the removal rate is between 80% and 82%, indicating that the denitrifying sludge cultivated with river sediments has a high-efficient denitrification performance in treating actual municipal wastewater.

# 4. Conclusion

The river sediments can be successfully cultivated into denitrifying sludge by adding river sediments to the reactor daily and reducing the HRT gradually. The cultivated sludge has a highly efficient denitrification performance that the removal rate of nitrate in municipal wastewater exceeds 80%. According to the results of RSM, the denitrifying sludge achieved the highest activity with the following conditions: $NO_3^-$-N 50 mg $l^{-1}$, HRT 6 h, and adding 6 ml river sediment to 1 l wastewater of reactor per day, and in this case the NRR reached 85.5%. The major denitrifying bacteria genera and the relative abundance in the cultivated denitrifying sludge were *Diaphorobacter* (33.82%) and *Paracoccus* (24.49%), showed by high-throughput sequencing.

Data accessibility. All data generated or analysed during this study are included in this paper; everyone could attempt a replication of this study according to the methods and data in this paper.
Authors' contributions. L.H. and J.L. designed the study, coordinated the study and draft the manuscript; Z.Z. and Q.S. carried out the data analyses; Y.L. and K.Z. collected field data. All authors gave final approval for publication.
Competing interests. The authors declare no competing interests.
Funding. The Major Science and Technology Program for Water Pollution Control and Treatment of China (nos. 2017zx07103-001) provided financial support for all authors in study.
Acknowledgements. The authors gratefully acknowledge the financial support from the Major Science and Technology Program for Water Pollution Control and Treatment of China.

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
