## [Reviewer comments · Royal Society Open Science]

Review History

RSOS-190304.R0 (Original submission)

Review form: Reviewer 1

Is the manuscript scientifically sound in its present form?

No

Are the interpretations and conclusions justified by the results?

No

Is the language acceptable?

Yes

Is it clear how to access all supporting data?

No

Do you have any ethical concerns with this paper?

No

Have you any concerns about statistical analyses in this paper?

I do not feel qualified to assess the statistics

Recommendation?

Reject

Comments to the Author(s)

The paper presented a method to cultivate a denitrifying sludge for treating wastewater. Unfortunately, the results in this paper do not present a novelty that could justify its publication. The reasons presented in the Introduction section are not supported by the references. In fact, the references cited address the issues related to obtain sludge able to remove both nitrogen and phosphorus and produce PHA. The present paper focused only in removal of nitrogen. There is much process that converts nitrogen and do not have issues to obtain an adequate sludge, e.g. the conventional nitrification-denitrification process.

Some information should be mentioned in the methods section. The number of deposit of sequences is required when using high-throughput sequencing data. The conditions of test of item 3.4 was not mentioned in the methods section as the details of RSA evaluation (item 3.2). Additionally, the formatting and spelling should be revised as some error was found (e. g. cement-water ratio, m3d, membrane 0.45 mm, ...).

Review form: Reviewer 2

Is the manuscript scientifically sound in its present form?

Yes

Are the interpretations and conclusions justified by the results?

Yes

Is the language acceptable?

Yes

Is it clear how to access all supporting data?

Yes

Do you have any ethical concerns with this paper?

No

Have you any concerns about statistical analyses in this paper?

I do not feel qualified to assess the statistics

Recommendation?

Accept with minor revision (please list in comments)

Comments to the Author(s)

Overall comments

- I enjoyed this article. This work focuses on developing a method to convert river sediment into denitrifying sludges for the treatment of real municipal wastewater. I would recommend it for acceptance after the points listed below are addressed. I hope these comments will be helpful.

Specific comments

- Introduction part: In my opinion, not all of the river sediments have the toxic or harmful substances because this is strongly related to the situation (or the degree of pollution) of the water environment. It would be better to mention this point in the Introduction section.
- Introduction part: It would be better to focus on the novelty of the sludge cultivation method in this study. I think the novelty is the procedure "the amount of sediment was gradually increased (page 2, line 27)". I recommend that the authors should describe well the reason why the authors applied this procedure.
- Page 1, line 18: a space should be inserted between "reactor" and "(SBR)", and "reactor" and "(daily)".
- Page 1, line 20: please replace "NO₃--N" by "NO₃⁻-N".
- Page 1, line 44: please insert a space before "However".
- Page 1, line 49: why is ammonium concentration of municipal wastewater so low?
- Page 3, Fig. 3: why did NRR decrease with an increase of SA from 6 to 8 mL? It might be expected that NRR increases with an increase of SA.
- In Abstract part: it would be better to describe the obtained results from Fig. 3 in more detail; for instance, "RSA analysis represents that nitrate removal was affected mainly by HRT, followed by SA" etc.
- Page 4 (Fig. 5): The legends of the figure are hard to see probably due to low image quality. Please revise them.
- In Reference part: please recheck this part. For instance, Reference No. 10, the authors forgot to write Journal or Book name.

Decision letter (RSOS-190304.R0)

03-Apr-2019

Dear Dr Li,

The editors assigned to your paper ("Cultivating river sediments into efficient denitrifying sludge for treating municipal wastewater") have now received comments from reviewers. We would like you to revise your paper in accordance with the referee and Associate Editor suggestions which can be found below (not including confidential reports to the Editor). Please note this decision does not guarantee eventual acceptance.

Please submit a copy of your revised paper before 26-Apr-2019. Please note that the revision deadline will expire at 00.00am on this date. If we do not hear from you within this time then it will be assumed that the paper has been withdrawn. In exceptional circumstances, extensions may be possible if agreed with the Editorial Office in advance. We do not allow multiple rounds of revision so we urge you to make every effort to fully address all of the comments at this stage. If deemed necessary by the Editors, your manuscript will be sent back to one or more of the original reviewers for assessment. If the original reviewers are not available, we may invite new reviewers.

When submitting your revised manuscript, you must respond to the comments made by the

referees and upload a file "Response to Referees" in "Section 6 - File Upload". Please use this to document how you have responded to the comments, and the adjustments you have made. In order to expedite the processing of the revised manuscript, please be as specific as possible in your response.

- Data accessibility

If you wish to submit your supporting data or code to Dryad (<http://datadryad.org/>), or modify your current submission to dryad, please use the following link:
<http://datadryad.org/submit?journalID=RSOS&manu=RSOS-190304>

- Competing interests

- Authors' contributions

- Acknowledgements

- Funding statement

Kind regards,

Andrew Dunn

on behalf of Prof R. Kerry Rowe (Subject Editor)

Associate Editor's comments:

There is a split opinion among the reviewers. We are giving you the benefit of the doubt to try and address the concerns raised by referee 1 in particular. You should fully respond to and incorporate the changes they request to make the paper potentially ready for publication. If you do not do so, or you cannot persuade the reviewer that the manuscript is ready for publication following revision, we will not be able to consider the manuscript further.

Comments to Author:

Reviewers' Comments to Author:

Reviewer: 1

Comments to the Author(s)

The paper presented a method to cultivate a denitrifying sludge for treating wastewater. Unfortunately, the results in this paper do not present a novelty that could justify its publication. The reasons presented in the Introduction section are not supported by the references. In fact, the references cited address the issues related to obtain sludge able to remove both nitrogen and phosphorus and produce PHA. The present paper focused only in removal of nitrogen. There is much process that converts nitrogen and do not have issues to obtain an adequate sludge, e.g. the conventional nitrification-denitrification process.

Some information should be mentioned in the methods section. The number of deposit of sequences is required when using high-throughput sequencing data. The conditions of test of item 3.4 was not mentioned in the methods section as the details of RSA evaluation (item 3.2). Additionally, the formatting and spelling should be revised as some error was found (e. g. cement-water ratio, m3d, membrane 0.45 mm, ...).

Reviewer: 2

Comments to the Author(s)

Overall comments

- I enjoyed this article. This work focuses on developing a method to convert river sediment into denitrifying sludges for the treatment of real municipal wastewater. I would recommend it for acceptance after the points listed below are addressed. I hope these comments will be helpful.

Specific comments

- Introduction part: In my opinion, not all of the river sediments have the toxic or harmful substances because this is strongly related to the situation (or the degree of pollution) of the water environment. It would be better to mention this point in the Introduction section.
- Introduction part: It would be better to focus on the novelty of the sludge cultivation method in this study. I think the novelty is the procedure “the amount of sediment was gradually increased (page 2, line 27)”. I recommend that the authors should describe well the reason why the authors applied this procedure.
- Page 1, line 18: a space should be inserted between “reactor” and “(SBR)”, and “reactor” and “(daily)”.
- Page 1, line 20: please replace “NO₃--N” by “NO₃⁻”.
- Page 1, line 44: please insert a space before “However”.
- Page 1, line 49: why is ammonium concentration of municipal wastewater so low?
- Page 3, Fig. 3: why did NRR decrease with an increase of SA from 6 to 8 mL? It might be expected that NRR increases with an increase of SA.
- In Abstract part: it would be better to describe the obtained results from Fig. 3 in more detail; for instance, “RSA analysis represents that nitrate removal was affected mainly by HRT, followed by SA” etc.
- Page 4 (Fig. 5): The legends of the figure are hard to see probably due to low image quality. Please revise them.
- In Reference part: please recheck this part. For instance, Reference No. 10, the authors forgot to write Journal or Book name.

Author's Response to Decision Letter for (RSOS-190304.R0)

See Appendix A.

RSOS-190304.R1 (Revision)

Review form: Reviewer 3

Is the manuscript scientifically sound in its present form?

Yes

Are the interpretations and conclusions justified by the results?

Yes

Is the language acceptable?

Yes

Do you have any ethical concerns with this paper?

No

Have you any concerns about statistical analyses in this paper?

No

Recommendation?

Accept with minor revision (please list in comments)

Comments to the Author(s)

The denitrification process would be generated nitrite and its unioned form free nitrous acid. It was reported that free nitrous acid caused inhibition to a broad of microbes (see Free nitrous acid-based nitrifying sludge treatment in a two-sludge system obtains high polyhydroxyalkanoates accumulation and satisfied biological nutrients removal. Bioresource Technology, 2019, 284: 16-24; Water Research. 2018, 145: 113-124). Please make a brief discussion on this point based on the references available.

Review form: Reviewer 4 (Zhaoji Zhang)**Is the manuscript scientifically sound in its present form?**

Yes

Are the interpretations and conclusions justified by the results?

Yes

Is the language acceptable?

Yes

Do you have any ethical concerns with this paper?

No

Have you any concerns about statistical analyses in this paper?

No

Recommendation?

Accept as is

Comments to the Author(s)

The paper (Ref.No.RSOS-190304.R1) presents a case study of enriching denitrifying bacteria using river sediments as inoculum. Results show the enrichment process is acceptable and effective. The topic is interesting, and this manuscript contributes some new knowledge on recycling of river sediments and biological nitrogen removal. The authors have made a detailed revision according to the reviewers. The manuscript is acceptance its current form.

Decision letter (RSOS-190304.R1)

13-Aug-2019

Dear Dr Li:

On behalf of the Editors, I am pleased to inform you that your Manuscript RSOS-190304.R1 entitled "Cultivating river sediments into efficient denitrifying sludge for treating municipal

wastewater" has been accepted for publication in Royal Society Open Science subject to minor revision in accordance with the referee suggestions. Please find the referees' comments at the end of this email.

The reviewers and Subject Editor have recommended publication, but also suggest some minor revisions to your manuscript. Therefore, I invite you to respond to the comments and revise your manuscript.

- Ethics statement

- Data accessibility

If you wish to submit your supporting data or code to Dryad (<http://datadryad.org/>), or modify your current submission to dryad, please use the following link:
<http://datadryad.org/submit?journalID=RSOS&manu=RSOS-190304.R1>

- Competing interests

- Authors' contributions

- Acknowledgements

- Funding statement

Because the schedule for publication is very tight, it is a condition of publication that you submit the revised version of your manuscript before 22-Aug-2019. Please note that the revision deadline will expire at 00.00am on this date. If you do not think you will be able to meet this date please let me know immediately.

on behalf of Prof R. Kerry Rowe (Subject Editor)
openscience@royalsociety.org

Associate Editor Comments to Author:

We have received two positive reports on your manuscript, however before we can formally accept your paper for publication, we would be grateful if you could address the final comment raised by Reviewer #1. Once you have revised your work as requested, and provided a response file to this comment, please resubmit your manuscript for further consideration.

Reviewer comments to Author:

Reviewer: 3

Comments to the Author(s)

The denitrification process would be generated nitrite and its unioned form free nitrous acid. It was reported that free nitrous acid caused inhibition to a broad of microbes (see Free nitrous acid-based nitrifying sludge treatment in a two-sludge system obtains high polyhydroxyalkanoates accumulation and satisfied biological nutrients removal. Bioresource Technology, 2019, 284: 16-24; Water Research. 2018, 145: 113-124). Please make a brief discussion on this point based on the references available.

Reviewer: 4

Comments to the Author(s)

The paper (Ref.No.RSOS-190304.R1) presents a case study of enriching denitrifying bacteria using river sediments as inoculum. Results show the enrichment process is acceptable and effective. The topic is interesting, and this manuscript contributes some new knowledge on recycling of river sediments and biological nitrogen removal. The authors have made a detailed revision according to the reviewers. The manuscript is acceptance its current form.

Author's Response to Decision Letter for (RSOS-190304.R1)

See Appendix B.

Decision letter (RSOS-190304.R2)

22-Aug-2019

Dear Dr Li,

I am pleased to inform you that your manuscript entitled "Cultivating river sediments into efficient denitrifying sludge for treating municipal wastewater" is now accepted for publication in Royal Society Open Science.

on behalf of Prof R. Kerry Rowe (Subject Editor)
openscience@royalsociety.org

Appendix A

Manuscript Details

Manuscript number : RSOS-190304

Title : Cultivating river sediments into efficient denitrifying sludge for treating municipal wastewater

Abstract : The river sediment contains a lot of pollutants in many cases, which needs to be treated appropriately for the restoration of water environments. In this study, a novel method was developed to convert river sediment into denitrifying sludge in a sequencing batch reactor (SBR). The river sediment was added into the reactor daily and the hydraulic retention time (HRT) of the reactor was gradually reduced from 8h to 4h. The reactor achieved in the NO_3^- -N removal efficiency of 85% with the NO_3^- -N removal rate of 0.27 kg N/(m³·d). Response Surface analysis represents that nitrate removal was affected mainly by HRT, followed by sediment addition. The denitrifying sludge achieved the highest activity with the following conditions: NO_3^- -N 50mg/L, HRT 6h, and adding 6 mL river sediments to 1L wastewater of reactor per day. As a result, the cultivated denitrifying sludge could removal 80% NO_3^- -N for real municipal wastewater, and the high-throughput sequence analysis indicated that major denitrifying bacteria genera and the relative abundance in the cultivated denitrifying sludge were *Diaphorobacter* (33.82%) and *Paracoccus* (24.49%). The river sediments cultivating method in this report can not only obtain denitrifying sludge but also make use of sediment resources, which has great application potential.

Dear editor:

Hope everything finds you well.

We would like to submit the revised manuscript and we wish it could be considered for publication in “RSOS”. We have made the changes according to the comments. The following is a detailed reply, and the revised manuscript has been submitted. We appreciate your consideration of our manuscript, Thank you.

If you have any questions, please contact us without hesitation at the address below.

Thank you and best regards.

Yours sincerely,

Liangang Hou, Jun Li, et al.

Corresponding author:

Name: Jun Li

E-mail: bjutlijun@sina.com

Reviewer: 1

Comment 1 : paper presented a method to cultivate a denitrifying sludge for treating wastewater. Unfortunately, the results in this paper do not present a novelty that could justify its publication. The reasons presented in the Introduction section are not supported by the references. In fact, the references cited address the issues related to obtain sludge able to remove both nitrogen and phosphorus and produce PHA. The present paper focused only in removal of nitrogen. There is much process that converts nitrogen and do not have issues to obtain an adequate sludge, e.g. the conventional nitrification-denitrification process.

Reply : Thank you for your comments. The paper focuses on developing a method that cultivate sediments to denitrifying sludge for treating wastewater, we think the novelty is the procedure of sediment cultivate methods that “the amount of sediment was gradually increased” and “the hydraulic retention time of the reactor was gradually reduced”. We think the method is suitable not only for the cultivation of denitrifying sludge, but also for other sludge in the wastewater treatment industry. In addition, although there are some processes that can obtain sludge, this sludge culture method can be used as a supplement to provide some help for some wastewater treatment stations, newly built wastewater treatment plants, etc. where there is no way to obtain sludge. We added new references to support our ideas in the introduction section, such as reference 10,14,15,18,24,29,30, etc.

Comment 2 : Some information should be mentioned in the methods section. The number of deposit of sequences is required when using high-throughput sequencing data. The conditions of test of item 3.4 was not mentioned in the methods section as the details of RSA evaluation (item 3.2).

Reply : I appreciate this suggestion. The number of deposit of sequences has been supplemented in the methods section “2.4 Community analysis”.

The conditions of test of item 3.4 and 3.1 are basically the same, so we think that the item 3.4 experimental setup part can be referred to 2.2, and without need to repeating.

The details of RSA evaluation (item 3.2) have been supplemented in the

methods section, named “2.5 Response Surface Analysis”.

Comment 3 : Additionally, the formatting and spelling should be revised as some error was found (e. g. cement-water ratio, m3d, membrane 0.45 mm, ...).

Reply : We have carefully checked the manuscript, the errors in the manuscript have been revised, such as “cement-water ratio” have been revised as “moisture content”; “membrane 0.45 mm” have been revised as “membrane 0.45 μm ”; “m3d” have been revised as “m³d”, etc.

Reviewer: 2

Comment 1 : - Introduction part: In my opinion, not all of the river sediments have the toxic or harmful substances because this is strongly related to the situation (or the degree of pollution) of the water environment. It would be better to mention this point in the Introduction section.

- Introduction part: It would be better to focus on the novelty of the sludge cultivation method in this study. I think the novelty is the procedure “the amount of sediment was gradually increased (page 2, line 27)”. I recommend that the authors should describe well the reason why the authors applied this procedure.

Reply : I appreciate this suggestion. We have already added this point in the introduction section, we added “in some rivers” in the sentence, and it shown as “and the toxic or harmful substances in some rivers” in the manuscript.

In present study, the amount of sediment was gradually increased to culture denitrification bacteria. The reason for conducting the procedure was explained as follows. First, the river sediment contained denitrification bacteria and other heterotrophic bacteria. If a large amount of river sediment was added into the reactor at once, other heterotrophic bacteria would compete with denitrification bacteria over carbon source. It might take a long time to enrich denitrification bacteria due to the shortage of carbon source. Second, in the beginning of the start-up of the denitrification reactor, the reactor only contained a small amount of denitrification bacteria. After a large amount of river sediment was added in the reactor, a large amount of bacteria would be dead under anoxic condition. Some intermediate product might inhibit the growth of

denitrification bacteria. Third, some river sediment would contain toxic pollutants. If the river sediment was gradually added to the reactor, the concentration of these toxic pollutants could be diluted. The enrichment of denitrification bacteria would be much easier under low toxic pollutants concentration.

Comment 2 : Page 1, line 18: a space should be inserted between “reactor” and “(SBR)”, and “reactor” and “(daily)”.

Page 1, line 20: please replace “NO₃--N” by “NO₃⁻-N”.

Page 1, line 44: please insert a space before “However”.

Reply : It has been corrected as required.

Comment 3 : - Page 1, line 49: why is ammonium concentration of municipal wastewater so low?

Reply : The municipal wastewater used in the experiment is taken from the effluent from the aeration tank of a wastewater treatment plant. The aeration tank mainly performs the nitrification reaction, and the ammonium was almost nitrated completely, so the ammonium concentration is relatively low.

Comment 4 : - Page 3, Fig. 3: why did NRR decrease with an increase of SA from 6 to 8 mL? It might be expected that NRR increases with an increase of SA.

Reply : Although we expect NRR increases with an increase of SA, but that's what we found in experiment, we speculated that after the addition of sediments, the bacteria in the reactor would be disturbed and new bacteria haven't grown yet, so there would be a period of adaptation.

Comment 5 : -In Abstract part: it would be better to describe the obtained results from Fig. 3 in more detail; for instance, “RSA analysis represents that nitrate removal was affected mainly by HRT, followed by SA” etc.

Reply : I appreciate this suggestion. We have already added this point in the abstract section.

Comment 6 : - Page 4 (Fig. 5): The legends of the figure are hard to see probably due to low image quality. Please revise them.

Reply : It has been corrected as required, Fig. 5 on the last page of the revised manuscript.

Comment 7 : - In Reference part: please recheck this part. For instance, Reference No. 10, the authors forgot to write Journal or Book name.

Reply: We have carefully checked all the references, and it has been corrected as required.

Appendix B

Manuscript number : RSOS-190304.R1

Title: Cultivating river sediments into efficient denitrifying sludge for treating municipal wastewater

Dear editor:

Hope everything finds you well.

We have made the changes according to the comments. The following is a detailed reply, and the revised manuscript has been submitted. We appreciate your consideration of our manuscript. If you have any questions, please do not hesitate to get in touch. Thank you and best regards.

Yours sincerely,

Liangang Hou, Jun Li, et al.

Reviewer3#

Comments to the Author(s)

The denitrification process would be generated nitrite and its unioned form free nitrous acid. It was reported that free nitrous acid caused inhibition to a broad of microbes (see Free nitrous acid-based nitrifying sludge treatment in a two-sludge system obtains high polyhydroxyalkanoates accumulation and satisfied biological nutrients removal. Bioresource Technology, 2019, 284: 16-24; Water Research. 2018, 145: 113-124). Please make a brief discussion on this point based on the references available.

Reply : I appreciate this suggestion. Because the concentration of nitrite in the bioreactor was low, we think there was very little free nitrous acid in the bioreactor, so the free nitrous acid has a little inhibitory effect on microbes. We have already added this point (and add two new references) in the manuscript, and it shown as “It was reported [references 40-41 in manuscript] that nitrite and its incorporation form free nitrous acid caused inhibition to a broad of microbes. Due to the influence of HRT in the cultivation stage, denitrification was incomplete and a small accumulation of NO_2^- -N was observed in this study. The concentration of nitrite in the bioreactor was low, and the highest accumulation concentration of NO_2^- -N was 6.33mg/L (Fig.2), so the concentration of free nitrous acid was very low, which has a little inhibitory effect on microbes.” in the manuscript.